# The Rollers' Offset Position Influence on the Counter-Roller Flow-Forming Process

Chengcheng Zhu [1,*], Fan Li [2], Yuanzhe Dong [1], Shengdun Zhao [2], Jingxiang Lv [1] and Dean Meng [3,*]

1   School of Construction Machinery, Chang'an University, Xi'an 710064, China
2   School of Mechanical Engineering, Xi'an Jiaotong University, Xi'an 710049, China
3   School of Automobile, Chang'an University, Xi'an 710064, China
*   Correspondence: ccz@chd.edu.cn (C.Z.); deanmeng@chd.edu.cn (D.M.);
    Tel.: +86-13772434546 (C.Z.); +86-18729595281 (D.M.)

**Abstract:** Background: The general counter-roller flow-forming (CRFF) process rarely considers the roller's offset position for the symmetric rollers. However, the rollers' offset position can regulate the tube shape, force, and other features. Studying the novel asymmetric CRFF process, which is the CRFF process with the rollers' offset position, is essential. Methods: The influence of the rollers' offset position, the tube blank thickness, thickness reduction on the material deformation, flow-forming force, final tube middle radius, and thickness in the CRFF process are studied using AA5052 aluminum tube experiments and numerical simulation. Result: The final tubes with three tube blank thicknesses, four thickness reduction, and four rollers' offset positions were obtained by the symmetric and asymmetric CRFF processes. Conclusions: AA5052 aluminum alloy tube can be made by the novel asymmetric CRFF process using a small rollers' offset position (−17.5–0%). Different rollers' positions could change the tube's middle radius. With negative rollers' offset position, the outer roller force is larger than the inner roller force. The force differences increase with the increase of tube blank thickness, the increase of thickness reduction, and the decrease of rollers' offset position. The asymmetric CRFF process helps design and construct large tube flow-forming equipment.

**Keywords:** counter-roller flow-forming; offset position; deformation; numerical simulation; experiment





## 1. Introduction

The large tubes utilized in the aerospace industry are complicated and expensive to produce by welding, mandrel flow-forming, and other traditional manufacturing processes. Thus, some novel manufacturing technologies, including the counter-roller flow-forming (CRFF)process, were exploited to solve this problem [1]. The CRFF process is a metal-forming method for large cylinder parts with many advantages (e.g., high qualities, material utilization) [2,3].

The traditional mandrel flow-forming process only has rollers on one side of the tube blank [4]. However, the CRFF process has a different working situation. Several couples of inner and outer rollers feed on the axial direction to form the rotation tube during the CRFF process (Figure 1a). Finally, the short thick tube becomes a long thin tube. The gap between the inner and outer rollers determines the final thickness of the tube. In ideal situations, the inner and outer rollers are often identical and precisely symmetrical. There are rare studies on the roller's offset position in the general CRFF process. However, the rollers' position difference usually occurs in the actual flow-forming equipment, which significantly changes the tube shape, flow-forming force, and other features. The rollers' position difference is the rollers' offset position when using the same rollers. Furthermore, we attempt to control the rollers' position to improve the CRFF flexibility and obtain various tubes with the same tube blanks. Therefore, the influence of the rollers' offset position on the CRFF process needs to be studied. This novel CRFF process in which we actively control rollers' offset position is called the asymmetrical CRFF process. The "tube thickness reduction ratio" is written as "thickness reduction" for short.

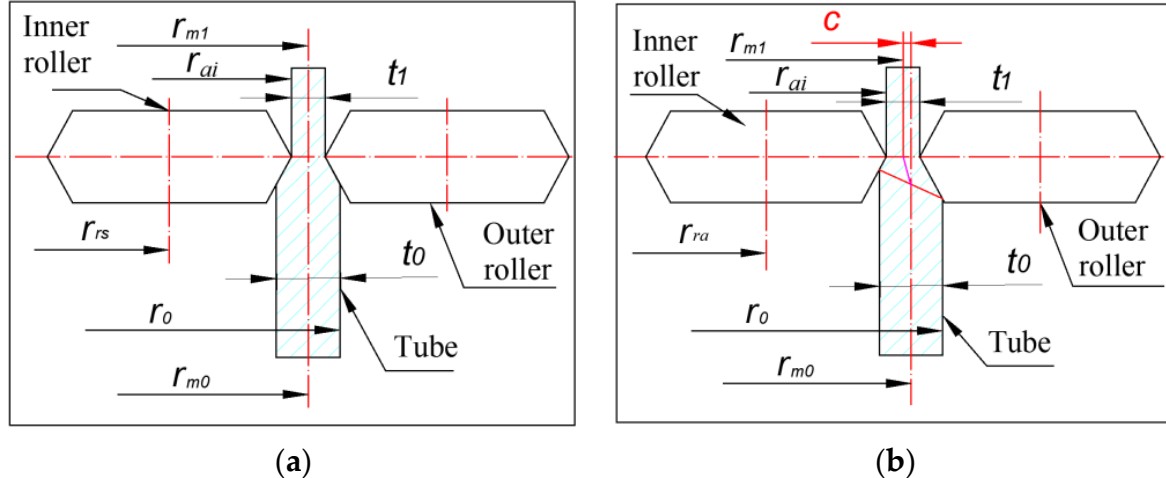

**Figure 1.** The comparison between different counter-roller flow-forming (CRFF) processes: (**a**) Symmetrical counter-roller flow-forming (SCRFF); (**b**) Asymmetrical counter-roller flow-forming (ACRFF).

The difference between the general symmetric counter-roller flow-forming (SCRFF) process and the asymmetric counter-roller flow-forming (ACRFF) process is shown in Figure 1b. The tube's inner and outer deformation regions are the same as in the general SCRFF process. Although the tube blank thickness $t_0$ is reduced to $t_1$ after the general SCRFF process, the middle radius of the final flow-formed tube $r_{m1}$ equals to the middle radius of the tube blank $r_{m0}$. The thickness reduction ratio is

$$\varnothing = -(t_1 - t_0)/t_0 = 1 - t_1/t_0, \tag{1}$$

where $\varnothing$ is the thickness reduction, $t_0$ is the tube blank thickness, and $t_1$ is the final tube thickness.

However, the inner and outer deformation regions are different in the ACRFF process. The middle radius of the final flow-formed tube $r_{m1}$ is different from the middle radius of the tube blank $r_{m0}$ in the ACRFF process. The radius difference is the rollers' offset position $c$. Thus, tubes of various middle radii with the same thickness can be obtained by changing the rollers' offset position.

$$c = r_{m1} - r_{m0} = (D_{m1} - D_{m0})/2, \tag{2}$$

$$c_r = c/t_0, \tag{3}$$

where $D_{m1}$ is the middle diameter of the final tube in the ACRFF process, $D_{m0}$ is the middle diameter of the tube blank, and $c_r$ is the relative rollers' offset position. In an ideal situation for simplifying the analysis, the tube just has plastic deformation during the CRFF process. Based on the experiment, the tube wall was assumed to keep it straight. The final tube inner surface radius $r_{ai}$ in the ACRFF process can be obtained by Equation (4). The tube blank middle radius $r_{m0}$ and the SCRFF final tube middle radius $r_{m1}$ is equal and calculated by Equation (5). The final tube middle radius in the ACRFF process can be obtained by Equation (6). The rollers' offset position also can be calculated by Equation (7)

$$r_{ai} = r_{ra} + R, \tag{4}$$

$$r_{m0} = r_0 - t_0/2, \tag{5}$$

$$r_{m1} = r_{ai} + t_1/2 = r_{ra} + R + (1 - \varnothing)t_0/2, \tag{6}$$

$$c = r_{m1} - r_{m0} = r_{ra} + R + \frac{(1 - \varnothing)t_0}{2} - \left(r_0 - \frac{t_0}{2}\right) = r_{ra} + R + \frac{(2 - \varnothing)t_0}{2} - r_0, \tag{7}$$

where $r_{ra}$ is the inner roller position in the ACRFF process, $R$ is the roller radius, and $r_0$ is the outer radius of the tube blank.

Many studies were noted on the general mandrel flow-forming process, including the working parameter optimization [5], microstructure evolution [6], tube size control [7], fracture prediction [8,9], rib formation [10], and parameter optimization [11]. Few studies on the asymmetric flow-forming method were noted because the general flow-forming process often has a cylinder mandrel. Furthermore, only a few studies were conducted on the rare CRFF process. The experiment device [12], primary material deformation mechanism [13], working parameters identification [14], and ring inner rib deformation in the CRFF process were obtained by the plastic theory, experiment, and numerical simulation method [15]. The rollers' offset position was also considered few in these studies.

The comparative study is on the asymmetric metal spinning process without a mandrel [16]. The preliminary studies of the asymmetric metal spinning processes are the path design [17], special material forming [18], and the influence of operational parameters [19]. Compared to the symmetrical flow-forming/metal-spinning process, the asymmetric metal-spinning process has a more complicated deformation mechanism, more operating parameters, and more unstable problems. Thus, studying the characteristics of the asymmetric flow-forming process before the process's practical applications is necessary.

This study is mainly about the rollers' offset position in the novel ACRFF process by numerical simulation and experimental methods. The AA5052 aluminum alloy tube was the research target. The influences of the three significant parameters are the tube blank thickness, thickness reduction, rollers' offset position on the material deformation, flow-forming force, the middle radius, and the final tube thickness. Various sizes of AA5052 aluminum alloy tubes can be made by the novel ACRFF process, which improves the flexibility of the CRFF process. The small inner roller force in the ACRFF helps improve the capability and simplify the structure of CRFF equipment. This study can improve the ACRFF deformation mechanism recognition and various large tube manufacturing methods.

## 2. Materials and Methods

### 2.1. Material Model

The widely used flow-forming material AA5052 aluminum alloy was selected for studying the CRFF process [20]. The density, Poisson's ratio, and Young's modulus of the AA5052 aluminum alloy were 2700 $kg/m^3$, 0.33, and 68 GPa, respectively. The AA5052 aluminum alloy has a strain rate sensitivity [21]. The hardening model of the AA5052 aluminum alloy was established by the uniaxial tensile test on an Instron universal testing machine. The shape and size of the tensile samples are shown in Figure 2a. The tensile speed during the test was 1.5 and 150 mm/min. The corresponding original strain rate was 0.001 and 0.1 $s^{-1}$. The strain rate of the sample during the test slightly decreased from 0.001 $s^{-1}$ to 0.00076 $s^{-1}$, so it is feasible to assume that the strain rate in the test is approximate invariant. The true strain-stress curve of the AA5052 aluminum alloy is shown in Figure 2b. The deformation speed influences the mechanical property of the AA5052 aluminum alloy. The yield strength improved, and the tensile strength and breaking elongation of the extension decreased with the enlargement of the tensile speed.

The AA5052 aluminum alloy elastoplastic constitutive model has been built by the Mises yield criterion, isotropic hardening model and linear elastic theory. The true strain-stress curve was fitted by the power function model [22]. The hardening curve's R-value and mean square error are bigger than 99% and less than 8 MPa, respectively.

$$\sigma = 426.3\varepsilon^{0.366}\dot{\varepsilon}^{-0.002} \tag{8}$$

where $\sigma$ is the stress/MPa, $\varepsilon$ is the strain, and $\dot{\varepsilon}$ is the strain rate/$s^{-1}$.

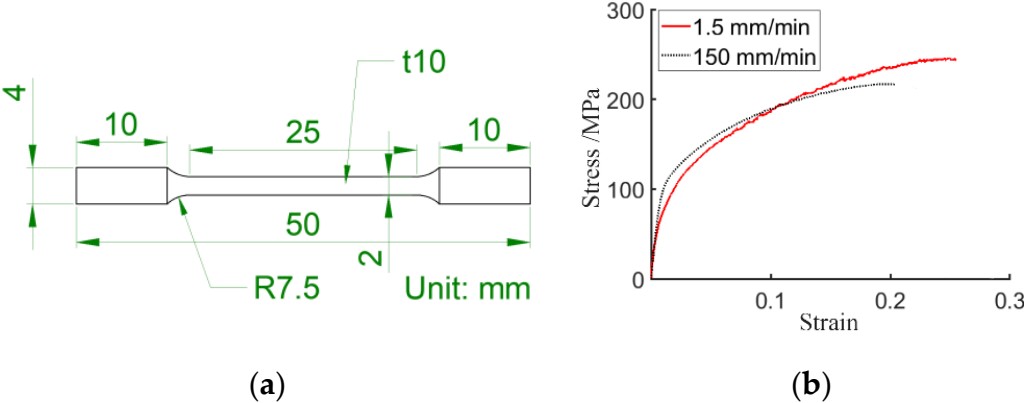

(a)    (b)

**Figure 2.** Tensile samples and strain-stress curve of AA5052 aluminum alloy: (**a**) Tensile sample; (**b**) Strain-stress curve.

### 2.2. Numerical Simulation Model

The 3D numerical simulation model is shown in Figure 3, similar to the experiment, based on the explicit dynamic method. Two pairs of inner and outer rollers with a diameter of 330 mm were used. The rollers and turntables are discrete rigid bodies meshed with R3D4 elements. The minimum size of the roller mesh was 0.7 mm. The tube blank material was AA5052 aluminum alloy whose hardening curve has a prominent saturation feature in significant deformation situations. There are many methods to extrapolate the hardening curve [23], but it is challenging to choose an appreciative one. The strain hardening curve used in the numerical simulationwas Equation (8). The maximum plastic strain used in this material model was 0.25 as the fracture strain in the tensile test. The stress would be an invariable value when the plastic strain was larger than 0.25.

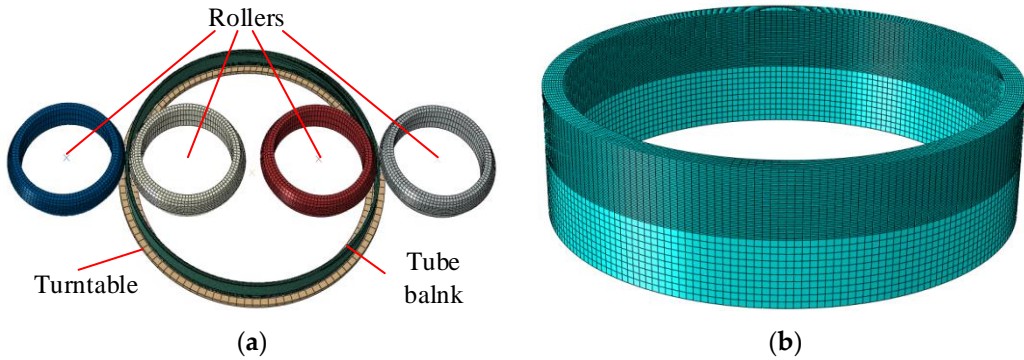

(a)    (b)

**Figure 3.** The numerical simulation model for the CRFF process: (**a**) Total model; (**b**) The mesh model of 720 mm diameter tube blank.

The outer diameter of the tube blank was 720 mm. The tube blank thickness was 10, 20, and 30 mm. The tube blank height was 50 and 200 mm. The tube blank was partitioned and meshed with the C3D8R element. The deformation region had a fine mesh of ~1 mm in the radial and axial directions. The bottom of the tube blank was fixed on the turntable, whose rotation speed was 0.67 r/s. The rollers' feed speed was 3 and 1 mm/s. The friction coefficient in the Coulomb friction model was 0.1, which is based on the AA5052 aluminum alloy ring compression test. The mass ratio is 1000 in the numerical model.

### 2.3. Experiment

The CRFF experiment was done on a numerical control CRFF device (Figure 4) at room temperature. The outer diameter and thickness of the AA5052 aluminum alloy tube blank were 720 and 10 mm, respectively. Since the experiment was based on the back-forward CRFF process, the tube blank bottom was fixed on the turntable. The turntable rotation

and roller feed speeds are 0.67 r/s and 1 mm/s, respectively. All rollers were at the same level and fed in the axial direction together as the synchronous CRFF method. The radial position of each roller was stationary. The tube blank's inner and outer surfaces were smeared with lubricating grease. All rollers were 330 mm in diameter and made of die steel. The thickness reduction of the final tube is 10, 20, 30, and 35%. The rollers' offset position is −0.25, −0.16, and −0.51 mm.

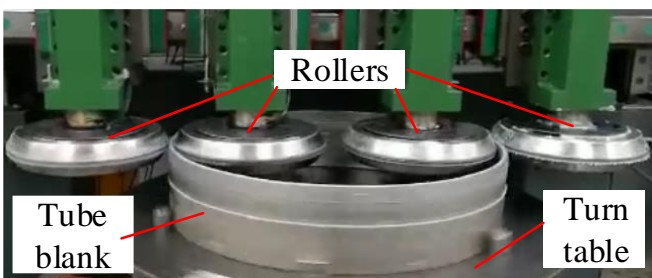

**Figure 4.** The experiment of the CRFF process.

### 3. Results

*3.1. General Result and Deformation Features*

The tube material has a particular deformation during the CRFF process. The material deformation features were studied on the final tubes of numerical simulation and experiment (Figure 5). The experiment and simulation's final tubes of CRFF processes matched well without fracture. Moreover, the general SCRFF tube had a slightly better size accuracy than the ACRFF tube.

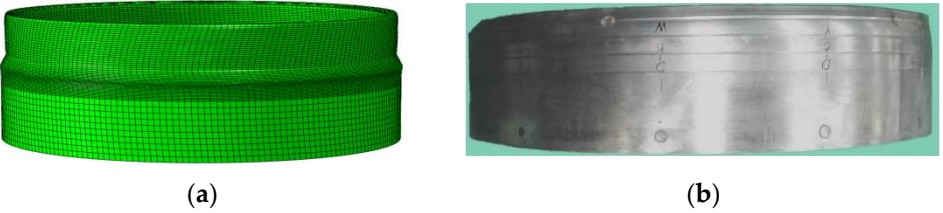

(**a**)  (**b**)

**Figure 5.** The final CRFF tubes: (**a**) Simulation; (**b**) Experiment.

The outer surface of the tube blank was plotted with rectangle nets with lengths of 14.2 and 13.4 mm in the axial and tangential directions, respectively (Figure 6a). The thickness reduction of the tube was 35% in the ACRFF process. The formed amount of the inner and outer tubes was 1.5 and 2 mm with 15% and 20% thickness reduction, respectively. The rollers' offset was −0.25 mm. The flow-formed tube had three regions: unformed, deformation, and formed (Figure 6b). The extended yellow line of B1, B2, and B3 could be regarded as the original net line because the material in the unformed region did not deform during the ACRFF process. The tube rotated to the right side in the experiment. The net line became curves from straight lines, and their position was at the back of the ideal position. Thus, some tangential deformation was noted in the ACRFF process. The average size of the nets in region A was 19.5 mm in the axial directions The net elongation ratio was ~37.2%, nearly the ideal flow-forming thickness reduction (35%). The tangential section of the flow-formed tube is shown in Figure 6c. The flow-formed tube top was bent inside. The stable deformation region had a similar thickness and vertical wall. Therefore, the main material deformation in the ACRFF process was in the axial and radial directions.

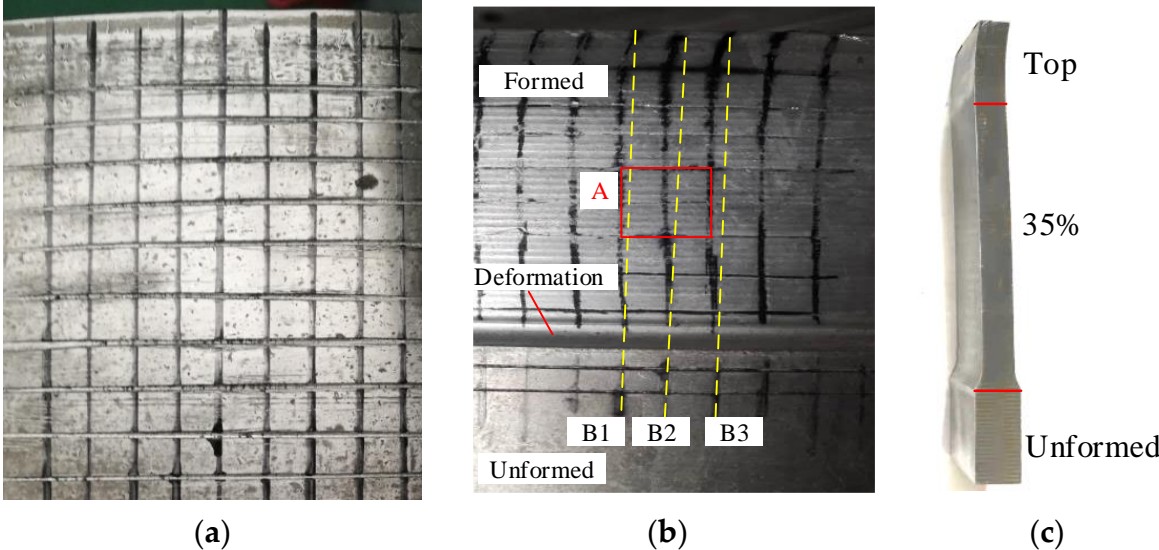

(a)　　　　　　　　　　　(b)　　　　　　　　　　　(c)

**Figure 6.** Tube deformation feature in the ACRFF process: (**a**) Nets on the tube blank; (**b**) Nets on the flow-formed tube (A is the sample of the deformation region; B1, B2 and B3 are Three lines in the axial direction); (**c**) Section of the flow-formed tube.

The numerical simulation tube also showed approximate deformation features. The tube stress nephogram in the ACRFF numerical simulation is shown in Figure 7a. The tube blank thickness is 10 mm. The tube thickness reduction is 35%, and the rollers' offset position is −0.25 mm, as used in the experiment. The deformation region and the tube bottom fixed on the turntable had large Mises stress of ~245.9 MPa. Figure 7b shows the final tube net like the experiment result: the tube net was elongated in the axial direction. The deformed net in the tube top distorts to the left side of the undeformed net on the tube bottom.

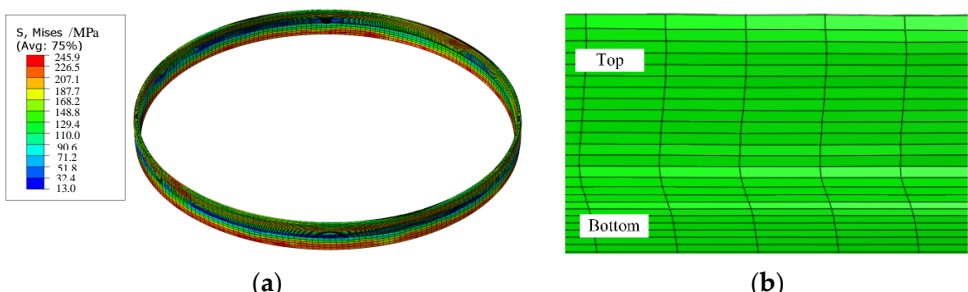

(**a**)　　　　　　　　　　　　　　　　　　(**b**)

**Figure 7.** The stress nephogram of the CRFF part: (**a**) Total model; (**b**) Final tube net.

*3.2. Tube Shape*

The tube's middle radius and thickness were chosen as significant parameters for studying the tube shape variation during the CRFF process. The middle radius and the thickness of the tube can be obtained by

$$r_{id} = (r_{ii} + r_{io})/2 \tag{9}$$

$$r_{re} = (r_{ri} + r_{ro})/2 \tag{10}$$

$$e_r = r_{re} - r_{id} \tag{11}$$

$$t_{id} = r_{io} - r_{ii} \tag{12}$$

$$t_{re} = r_{ro} - r_{ri} \tag{13}$$

$$e_t = t_{re} - t_{id} \tag{14}$$

where $r_{id}$, $r_{ii}$, and $r_{io}$ is the middle, inner, and outer radius of the final tube in the ideal situation, respectively; $r_{re}$, $r_{ri}$, and $r_{ro}$ is the middle, inner, and outer radius of the final tube in the real situation, respectively; and $e_r$ is the tube radius difference. The $t_{id}$ and $t_{re}$ is the tube thickness in ideal and real situations, respectively. The $e_t$ is the tube thickness difference.

The thickness of the tube blank was 10 mm. The stage tube made by the ACRFF process with 10, 20, and 30% ideal thickness reduction is shown in Figure 8. The final tube shape in the numerical simulation and the experiment are similar. The final tube shape values are shown in Table 1. In the 10% thickness reduction ACRFF process, the tube form depth difference and the rollers' offset position are −0.32 and −0.16 mm, respectively. In the 20% thickness reduction ACRFF process, the tube form depth difference and the rollers' offset position are −0.32 and −0.16 mm, respectively. In the 30% thickness reduction ACRFF process, the form depth difference and the rollers' offset position are −1.02 and −0.51 mm, respectively. The middle radius of the tube blank and the final tube in the general SCRFF process is 355 mm. The rollers' offset position changed the middle radii of the final tubes to 354.84 and 354.49 mm. Therefore, the middle radius of the CRFF tube depends on the rollers' offset position.

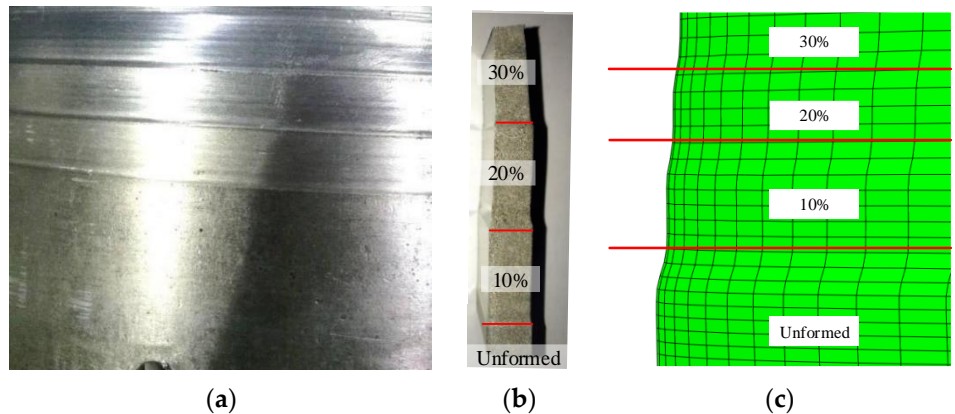

**Figure 8.** Tube shape with different ACRFF parameters: (**a**) Stage tube of the experiment; (**b**) Section of the stage tube; (**c**) Stage tube of Simulation.

**Table 1.** The shape parameters of the different ACRFF tubes.

| Item | Ideal Thickness/mm | Real Thickness/mm | Thickness Difference/mm | Real Offset Position/mm | Ideal Middle Radius/mm | Real Middle Radius/mm | Radius Difference/mm |
|---|---|---|---|---|---|---|---|
| | 9 | 9.08 | 0.08 | −0.16 | 354.80 | 354.84 | 0.04 |
| Experiment | 8 | 7.96 | −0.04 | −0.16 | 354.80 | 354.84 | 0.04 |
| | 7 | 6.86 | −0.14 | −0.51 | 354.50 | 354.49 | −0.01 |
| Simulation | 7 | 6.46 | −0.54 | −0.50 | 354.50 | 354.76 | 0.26 |

The tube shape parameters in the ACRFF numerical simulation are also shown in Table 1. The experiment result and simulation match the ideal tube thickness and middle radius well. The final tube thickness and middle radii are 6.46 mm and 354.76 mm, respectively, in the simulation and 6.86 mm and 354.49 mm, respectively, in the experiment. The tube thickness and radius differences are approximately −0.40 mm and 0.27 mm, respectively. Therefore, the simulation model is highly accurate and acceptable.

### 3.3. Flow-Forming Force

The inner and outer rollers have their force, generally called flow-forming force, during the CRFF process. The inner roller total force ($F_{i,\text{total}}$) is the resultant force of the inner roller radial force ($F_{i,r}$), tangential force ($F_{i,t}$), and axial force ($F_{i,a}$) as shown in Equation (15). The total force ($F_{o,\text{total}}$), radial force ($F_{o,r}$), tangential force ($F_{o,t}$), and axial force ($F_{o,a}$) of the outer roller also have the same relationship (Equation (16)). The force difference between the

outer and inner roller $\Delta F$ is calculated by Equation (17). The mean force of the outer and inner roller $F_m$ is obtained by Equation (18).

$$F_{i,total} = \sqrt{F_{i,r}^2 + F_{i,t}^2 + F_{i,a}^2} \tag{15}$$

$$F_{o,total} = \sqrt{F_{o,r}^2 + F_{o,t}^2 + F_{o,a}^2} \tag{16}$$

$$\Delta F = F_o - F_i \tag{17}$$

$$F_m = (F_o + F_i)/2 \tag{18}$$

where $F_o$ and $F_i$ are the outer and inner roller forces, respectively.

The flow-forming forces of the inner and outer rollers in different CRFF processes are shown in Figure 9. The inner and outer roller forces happen simultaneously in the SCRFF process for similar deformations. The outer roller force happens before the inner force for the different deformations in the ACRFF process. Similarities exist between the general SCRFF and the ACRFF forces. (1) All roller forces have similar evolution laws. The roller force increases at the beginning of the CRFF process and then becomes stable. (2) The relationship among tangential, axial, and radial forces is approximate. The most significant force is the radial force, and the tangential force is the most minor. The scale ratios of radial, tangential, and axial forces in the general SCRFF and ACRFF processes are 1:0.030:0.234 and 1:0.027:0.216.

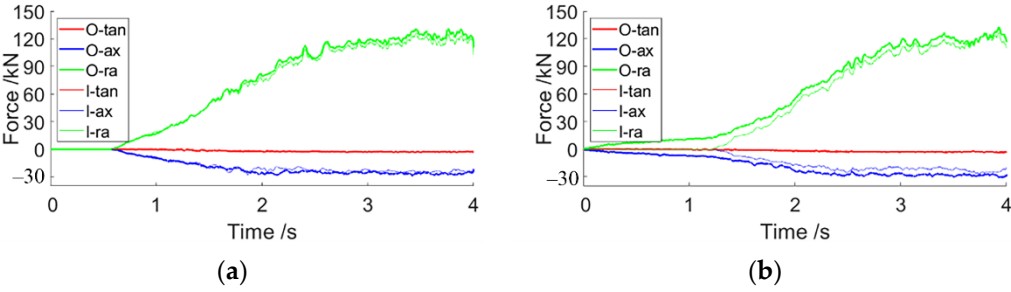

**Figure 9.** The flow-forming forces of inner roller and outer rollers with 35% thickness reduction within 4 s: (**a**) The SCRFF process; (**b**) The ACRFF process with −0.25 mm rollers' offset position. (O-tan: the outer roller tangential force; O-ax: the outer roller axial force; O-ra: the outer roller radial force; I-tan: the inner roller tangential force; I-ax: the inner roller axial force; I-ra: the inner roller radial force).

The force values in the SCRFF and ACRFF processes are different. Furthermore, the outer and inner roller forces have different relationships in the SCRFF and ACRFF processes. The outer roller radial force $F_{o,r}$ is similar to the $F_{i,r}$, with a difference of ~8 kN in the SCRFF process. The roller force difference $\Delta F$ can be as large as 96.6 kN when the tube thickness is 30 mm in the ACRFF process. The influence of rollers' offset position and other parameters on the flow-forming force will be discussed in the following sections.

The strain gauge sensor test device obtained the flow-forming force in the experiment. The experiment and the numerical simulation CRFF forces are compared when the tube blank has 720 mm diameter, 10 mm thickness, and 10% thickness reduction. The $F_{o,r}$ and $F_{o,a}$ in the experiment are 16.8 and 4.2 kN, respectively. The $F_{o,r}$ and $F_{o,a}$ in the simulation are ~23.0 and 4.7 kN, respectively. The $F_{o,r}$ and $F_{o,a}$ differences between the experiment and simulation are ~36.9% and 11.9%, respectively. The simulation model is acceptable.

## 4. Discussion

The influence of the tube blank thickness, rollers' offset position, and tube thickness reduction on the SCRFF and the ACRFF processes are discussed.

### 4.1. The Tube Blank Thickness

#### 4.1.1. Tube Deformation

The tube blank thickness was 10, 20, and 30 mm. The tube thickness reduction in the CRFF process was 35%. The rollers' offset position was −1.75, −3.5, and −5.25 mm in the 10, 20, and 30 mm thick tube blank ACRFF process, respectively. The stress nephograms of the tube deformation region during the CRFF processes are shown in Figure 10. Since the Equivalent plastic strain (PEEQ) in all numerical simulations is larger than 0.25, the stress will reach the sizeable invariant value in the material model. Furthermore, there are stress concentration phenomena in the contact area between the roller and the tube blank for large deformation. The maximum stress of the tube blank in all CRFF processes is ~240 MPa. The largest PEEQ value in the contact area in the numerical simulation is 1.7, which is bigger than the equivalent strain of 0.5 in the theoretical model. The reason is the asymmetric deformation and cyclic loading in the ACRFF process. The ACRFF tube has different deformation features from the general SCRFF tube. The forming region varies between the inner and outer tubes in the ACRFF process. The outer of the tube has a larger forming region and average stress than the inner. The tube net changes between the inner and outer roller gaps. The tube net in the general SCRFF process has symmetric deformation with a peak point in the middle of the tube. The peak point of the net in the ACRFF process is adjacent to the outer roller. Thus, the material flows faster in the tube outside than in the tube inside and causes the tube radius and flow-forming force variation.

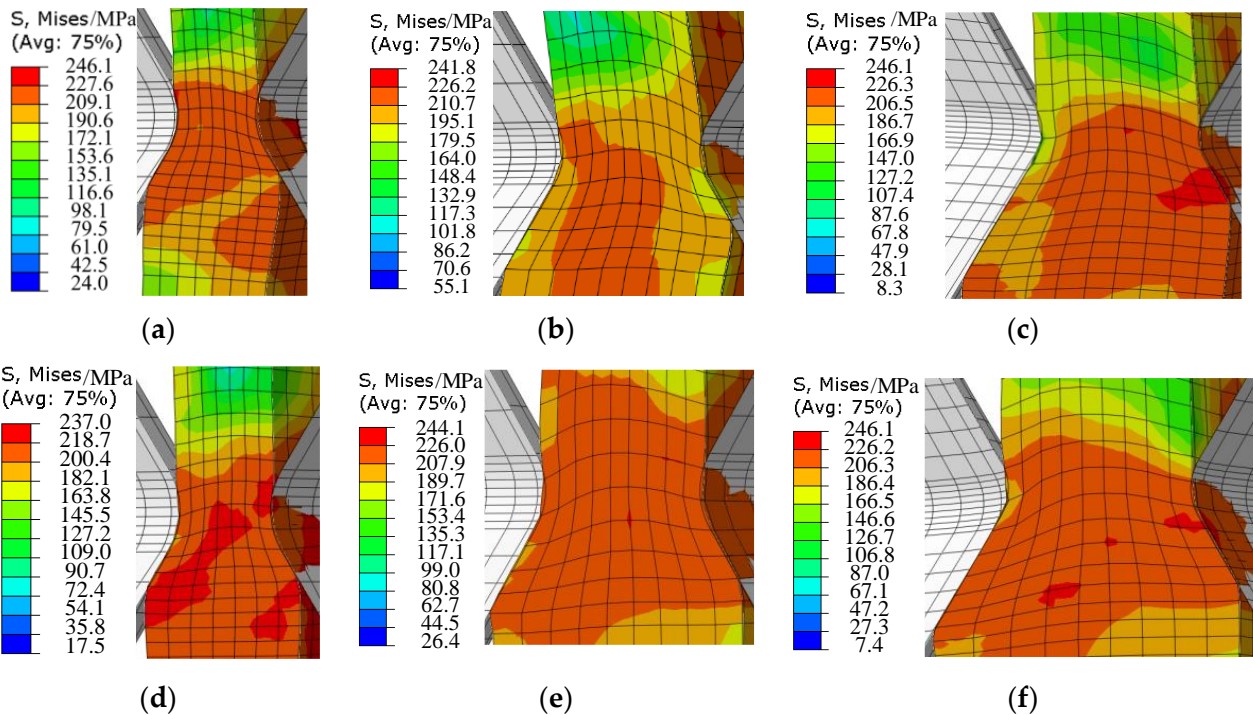

**Figure 10.** The stress nephograms of the tube deformation region in the CRFF process: (**a**) 10 mm thickness, ACRFF; (**b**) 20 mm thickness, ACRFF; (**c**) 30 mm thickness, ACRFF; (**d**) 10 mm thickness, SCRFF; (**e**) 20 mm thickness, SCRFF; (**f**) 30 mm thickness, SCRFF.

The contact region between the roller and the tube increases with tube blank thickness. In Figure 10c, the outer roller contact line is 23.29 mm and ~2.84 times the inner roller contact line length in the 30 mm thick-tube ACRFF process. The outer roller contact line length in the 10 mm thick-tube ACRFF process is 0.28 times that in the 30 mm thick-tube ACRFF process. The primary reason is that more material deforms in the thick tube than in the thin tube with similar thickness reduction. The thicker tube has larger stiffness and is harder to form. Therefore, the thick tube needs a large deformation region and roller contact line to provide a sizeable forming force.

### 4.1.2. Flow Forming Force

The flow-forming forces in each CRFF process are shown in Figure 11. All flow-forming forces increase with the tube blank thickness. When the tube thickness is 10, 20, and 30 mm, the $F_{o,\text{total}}$ in the general SCRFF processes is 152.7, 243.1, and 313.9 kN, respectively. The outer roller force in the ACRFF process increases faster than in the SCRFF process. The difference of the $F_{o,\text{total}}$ between the SCRFF and ACRFF processes are 8.1 and 57.5 kN when tube blank thickness is 10 and 30 mm, respectively. The roller force difference ($\Delta F$) is slight in the SCRFF process without tube thickness influence. However, the $\Delta F$ is significant in the ACRFF process. The difference between the $F_{i,\text{total}}$ in the SCRFF and ACRFF processes are 4.3 and $-38.6$ kN when tube thickness is 10 and 30 mm, respectively. All flow-forming forces have similar evolution laws. Therefore, the larger tube blank thickness will arouse the $F_o$ and the $\Delta F$. The mean force ($F_m$) in the ACRFF process is a little larger than that in the SCRFF process. When the tube blank thickness is 10, 20, and 30 mm, the total mean force differences are 6.5, 12.4, and 6.5 kN, respectively.

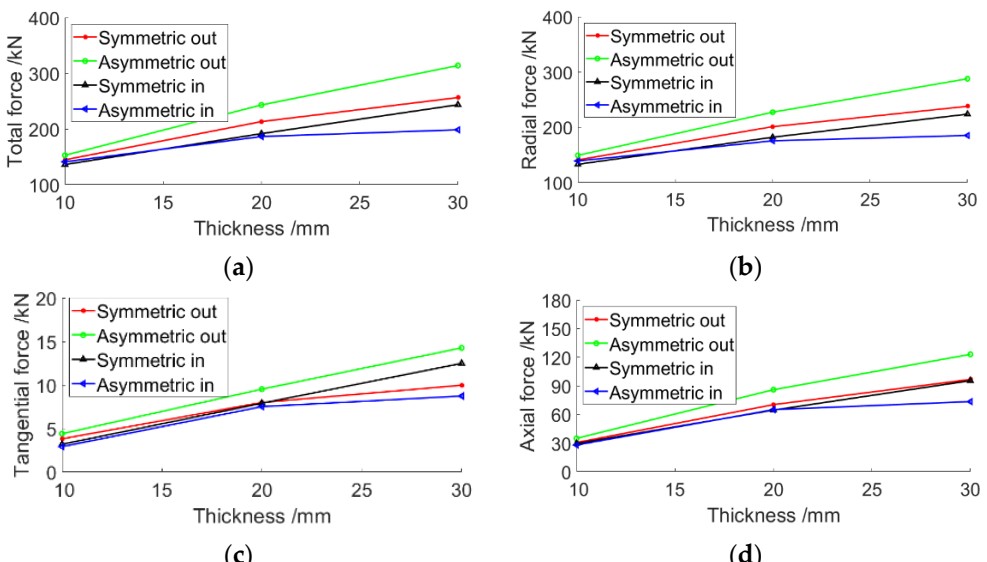

**Figure 11.** The influence of tube blank thickness on flow-forming force: (**a**) Total force; (**b**) Radial force; (**c**) Tangential force; (**d**) Axial force. (out: outer roller; in: inner roller.).

The scale ratios of the radial force $F_{o,r}$, $F_{o,t}$ and $F_{o,a}$ in the SCRFF processes are 1:0.030:0.234, 1:0.042:0.378, and 1:0.050:0.428 when the tube blank thickness is 10, 20, and 30 mm, respectively. When tube thickness is 10, 20, and 30 mm, the scale ratios of radial force $F_{o,r}$, $F_{o,t}$, and $F_{o,a}$ in the ACRFF processes are 1:0.027:0.216, 1:0.040:0.349, and 1:0.042:0.406, respectively. The tangential and axial forces will increase faster than the radial force with the enlargement of the tube thickness. Furthermore, the outer roller tangential and axial forces have a larger scale in the total flow-forming forces in the SCRFF process than in the ACRFF process.

### 4.1.3. Tube Shape

The influence of the tube blank thickness on the final tube shape is shown in Figure 12. The middle radius decrease with the enlargement of the tube thickness. The middle radius differences between the tubes in the ACRFF and the SCRFF processes are $-1.25$, $-2.45$, and $-3.94$ mm when the tube blank thickness is 10, 20, and 30 mm, respectively. The actual middle radius matched the ideal middle radius well in both SCRFF and ACRFF processes. The maximum relative middle radius differences in the SCRFF and ACRFF processes are 0.43% and 0.06%, respectively. Therefore, the negative rollers' offset position leads to a bit of expansion of the tube radius. Figure 12b shows the ideal and final tube thickness. The thickness of the final tubes of the SCRFF and the ACRFF processes is slightly larger than

the ideal thickness without considering the elastic deformation, with a relative difference of ~5.6%. The final tube thickness in the SCRFF and the ACRFF processes is 19.97 and 19.84 mm, respectively, when the tube thickness is 30 mm. The ideal final tube thickness is 19.5 mm. Thus, the ACRFF process could obtain a high-quality thin wall tube.

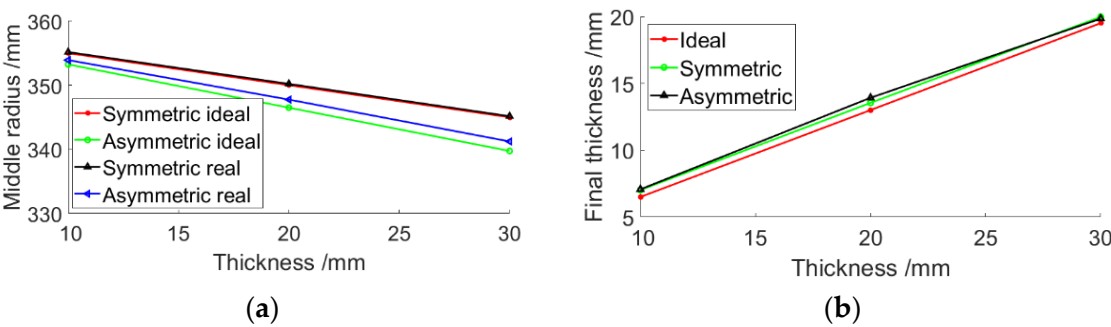

**Figure 12.** The influence of the tube blank thickness on the final tube shape: (**a**) Middle radius; (**b**) Final thickness. (ideal: ideal situation; real: real situation.).

### 4.2. Offset Position

#### 4.2.1. Tube Deformation

The thickness of the tube blank and the final tube was 30 and 19.5 mm, respectively. The tube thickness reduction $\varnothing$ was 35%. The rollers' offset position $c$ was $-5.25$, $-3$, $-0.75$, and 0 mm. The rollers' relative offset position $c_r$ was $-17.5\%$, $-10\%$, $-2.5\%$, and 0%. The CRFF process is the SCRFF when the rollers' offset position $c$ is 0%. Otherwise, the process is the ACRFF. Since the rollers' offset value is in the negative direction in this study, its offset value increases with its absolute value decreasing. The tube sections of the deformation region with different offset positions are shown in Figure 13. The rollers' offset position influences the deformation region. The mesh twists to the outside direction (left direction) with the value of the rollers' offset position decreasing. The contact line between the roller and the tube blank in Figure 13 is selected as the research parameter in the CRFF process (Table 2). The outer roller contact line length decreases with the increase of rollers' offset position. The inner roller contact line length has the opposite evolution law. The difference between the outer and inner roller contact line length decreases from 15.08 to 3.69 mm when the rollers' offset position value increases from $-17.5$ to 0%. The contact line length difference decreases with the increase of the rollers' offset position when the rollers' offset position is negative. The inner and outer rollers have different deformation because the contact line length difference is not zero in the CRFF process. A little special rollers' offset position may adjust the inner and outer roller deformations to be a real symmetry to improve tube stability.

This evolution law of contact line length directly changes the flow-forming force. In the CRFF process, the contact line length is dominated by many working parameters (e.g., thickness reduction, feed ratio, roller shape, etc.). In this study, all parameters except the rollers' offset position are constant values. Thus, the contact line length variation is due to the changing of the roller's position. The rollers' offset position decreases the tube's outer deformation region and increases the tube's inner deformation region. The roller contact line and flow-forming force will then change. The asymmetric deformation of the tube in the ACRFF leads the tube pressure and wall to deflect to one side. This deflection of the tube would further increase the asymmetric deformation.

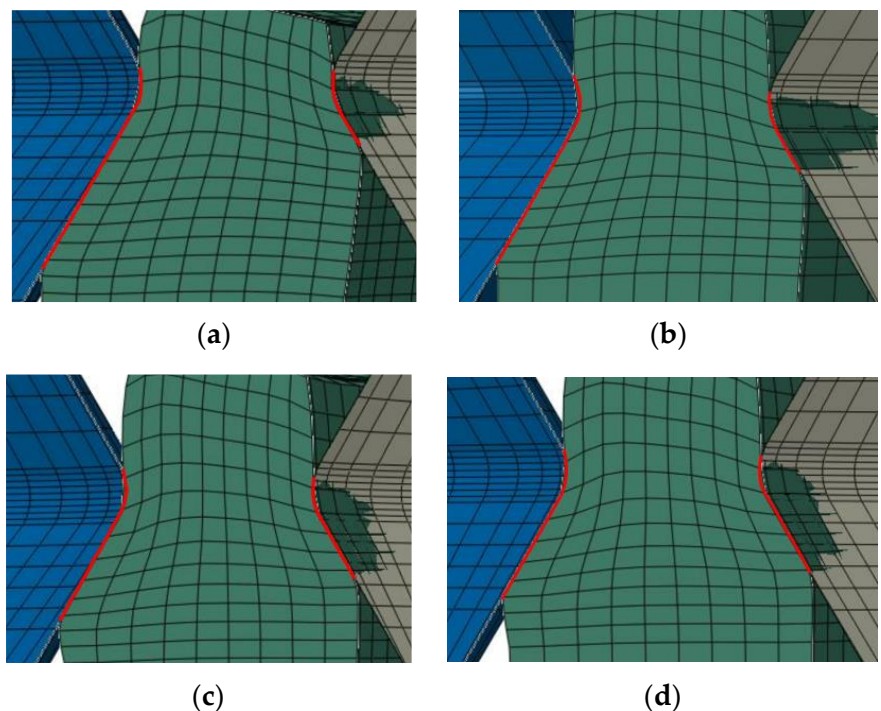

**Figure 13.** The influence of the rollers' offset position on the tube deformation: (**a**) −17.5%; (**b**) −10%; (**c**) −2.5%; (**d**) 0%.

**Table 2.** The length of the contact line between the roller and the tube.

| Relative Offset Position/% | −17.5 | −10 | −2.5 | 0 |
|---|---|---|---|---|
| Inner length/mm | 8.21 | 8.82 | 11.52 | 13.09 |
| Outer length/mm | 23.29 | 21.84 | 17.50 | 16.78 |
| Length ratio | 2.84 | 2.48 | 1.52 | 1.28 |
| Difference/kN | 15.08 | 13.02 | 5.98 | 3.69 |
| Relative difference/% | 65% | 60% | 34% | 22% |

4.2.2. Flow-Forming Force

The flow-forming forces of the inner and outer rollers are shown in Figure 14. The CRFF process has different results with different rollers' offset positions. The ACRFF process is stable with a slight negative rollers' offset position. All outer roller forces decrease with the increase of the rollers' offset position. For instance, $F_{o,\text{total}}$ decreases 57.5 kN while the relative offset position value increases 17.5% (from −5.25 mm to 0 mm). The inner roller force has the opposite variation law. The $F_{i,\text{total}}$ increases 45.0 kN at the same time. Thus, the outer roller force variation is slightly more significant than the inner roller force variation.

The $F_m$ is nearly constant at the different rollers' offset position conditions. The $F_{m,\text{total}}$, mean radial force ($F_{m,r}$), mean tangential force ($F_{m,t}$), and mean axial force ($F_{m,a}$) are 250.9, 232.6, 11.0, and 94.6 kN, respectively. The $\Delta F$ decreases with the increase of the rollers' offset position. The tangential force has a slight difference in evolution laws from other forces. The $F_{o,t}$ is smaller than the $F_{i,t}$ at 0% offset positionin the SCRFF process. Because the inner roller has a longer circumference contact line than the outer roller in the SCRFF process. Reducing the inner roller force is beneficial because the inner roller structure is more complicated than the outer roller structure in the CRFF process. Therefore, the ACRFF process with negative rollers' offset position helps reduce the designing and construction difficulty and cost of large tube CRFF equipment.

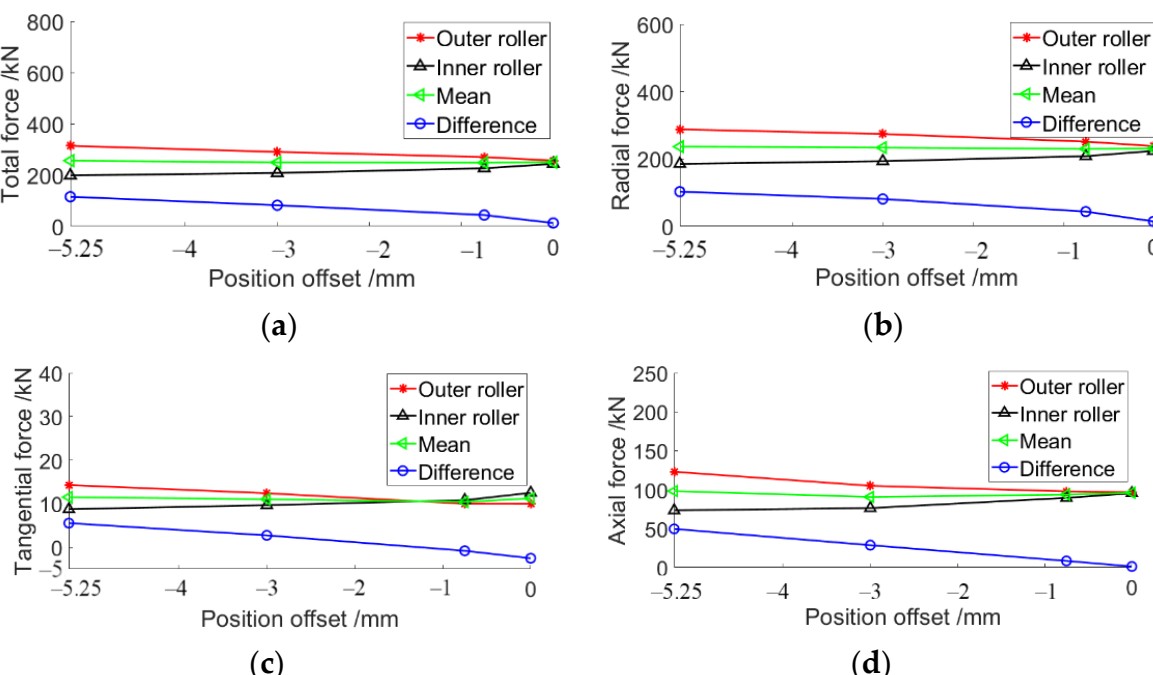

**Figure 14.** The influence of the rollers' offset position on the flow-forming force: (**a**) Total force; (**b**) Radial force; (**c**) Tangential force; (**d**) Axial force. (Mean: mean force; Difference: force difference).

### 4.2.3. Tube Shape

The middle radius and thickness of the final tube in the ideal and real situations are shown in Figure 15. The ideal and real middle radii are approximate. Since the rollers' offset value is in the negative direction in this study, its offset value increases with its absolute value decreasing. The middle radius increases with the increase of the rollers' offset position value in this situation. When the roller's offset position is negative, the middle radius increases with the absolute value decrease of the rollers' offset position. The actual tube thickness is near a constant value of 19.8 mm. Compared to the 19.5 mm thickness of the ideal final tube, the actual tube thickness's absolute difference and relative difference are about 0.3 mm and 1.52%, respectively. The ideal tube middle radius and thickness were calculated without the elastic deformation as general bulk forming analysis. Thus, the constant thickness difference could be regarded as the combination of elastic deformation and system differences. The thickness difference is a small constant value, and parameter compensation can reduce its influence. The tube radius can be adjusted by the rollers' offset position over a small range for a special-size tube.

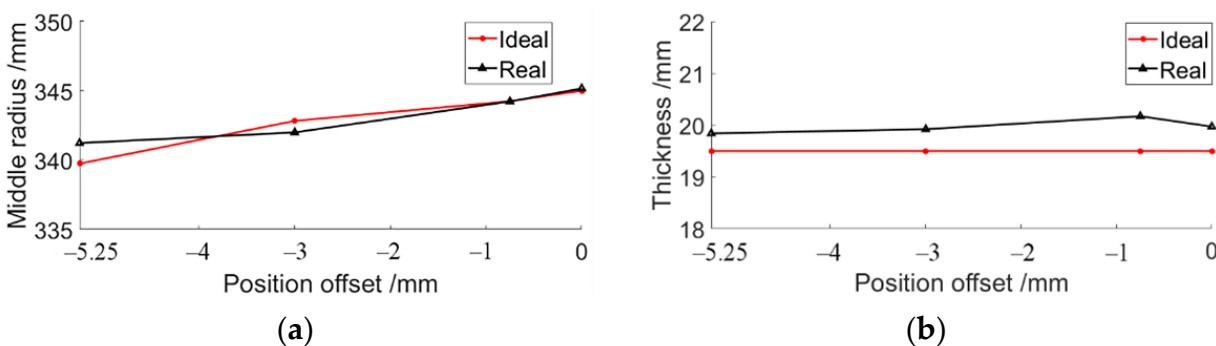

**Figure 15.** The influence of the rollers' offset position on the final tube shape: (**a**) Middle radius; (**b**) Thickness.

### 4.3. Thickness Reduction

#### 4.3.1. Tube Deformation

The tube blank thickness used in the SCRFF and ACRFF processes was 30 mm. The thickness reduction of the tube blank was 10, 20, 30, and 35%. The rollers' relative offset position $c_r$ was −half the thickness reduction in the ACRFF process that inner rollers just provide support at the beginning. The stress nephograms of the tube deformation region with different thickness reductions in ACRFF processes are shown in Figure 16. The maximum stress is ~240 MPa. The high-stress region area increases with the thickness reduction. Furthermore, the inner deformation region area and stress of the tube enlarges with the increase of thickness reduction in the ACRFF process. The stress of the inner tube contact region is ~188 and 224 MPa in the 10% and 35% thickness reduction ACRFF processes, respectively. The inner roller contact line length ratio in 35% and 10% thickness reduction is 4.19. The ratio of the outer roller contact line length is 2.53. Thus, the inner roller contact line length increases faster than the outer contact line length.

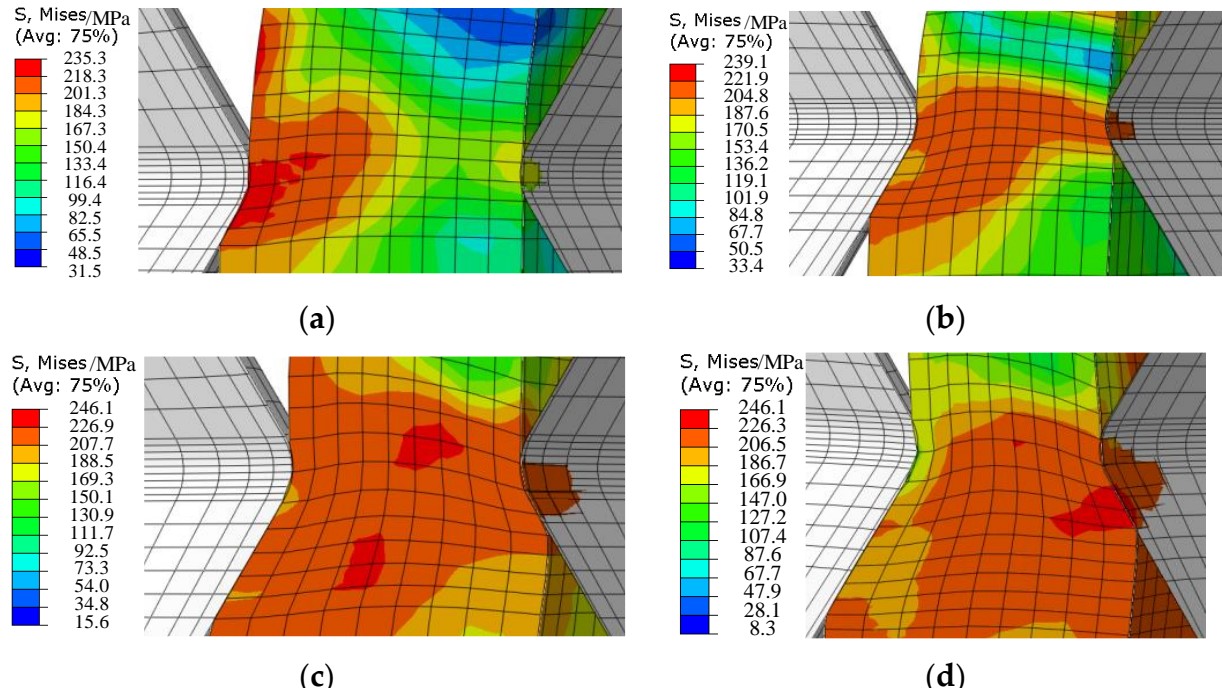

**Figure 16.** The stress nephograms of the tube deformation region with different thickness reduction in ACRFF processes: (**a**) 10%; (**b**) 20%; (**c**) 30%; (**d**) 35%.

The tube deformation in the ACRFF and SCRFF processes is quite different when the tube blank thickness is small. Large stress and approximate deformation in the 10% thickness reduction SCRFF process are observed in both the inner and outer of the tube (Figure 17). The large thickness reduction flow-forming process shows prominent plastic deformation in all tube deformation regions (Figure 10). However, only the outer of the tube has obvious deformation in the 10% thickness reduction ACRFF process with 30 mm thick tube. The 30 mm thick tube has a higher stiffness than the 10 mm thick tube. The whole roller gap region does not have any considerable plastic deformations. Thus, the inner roller has little influence on the small thickness reduction ACRFF process but supports the tube.

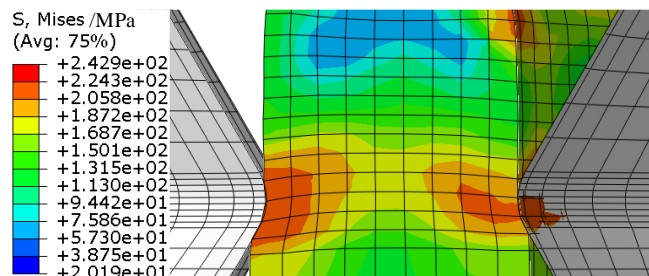

**Figure 17.** The stress nephogram of the tube deformation region in 10% thickness reduction SCRFF process.

### 4.3.2. Flow-Forming Force

As a fundamental working parameter, the thickness reduction significantly influences the CRFF process. The flow-forming forces of the inner and outer rollers in the SCRFF and the ACRFF process are shown in Figure 18. All flow-forming forces enlarge with the increase of thickness reduction as the general flow-forming process [24]. When the tube thickness reduction increases from 10% to 35%, the $F_{o,\text{total}}$ increases from 131.5 to 313.9 kN. The $F_o$ is much larger than the $F_i$ in the ACRFF process. The $\Delta F_{\text{total}}$ in the ACRFF processes are larger than 96 kN. The roller force in the SCRFF process is between the outer and inner roller forces in the ACRFF process. The outer roller force is approximate to the inner roller force in the SCRFF process except for the tangential force. The $F_{i,t}$ is larger than the $F_{o,t}$ in the SCRFF process. The $\Delta F_t$ value in SCRFF processes increase from $-0.57$ kN to $-2.52$ kN while thickness reduction increases from 10% to 35%. The $F_{m,\text{total}}$ in the ACRFF process can be larger 6 kN than that in the SCRFF process, except for the 10%. The difference in the mean total force between the ACRFF and SCRFF processes becomes larger when the thickness reduction increases.

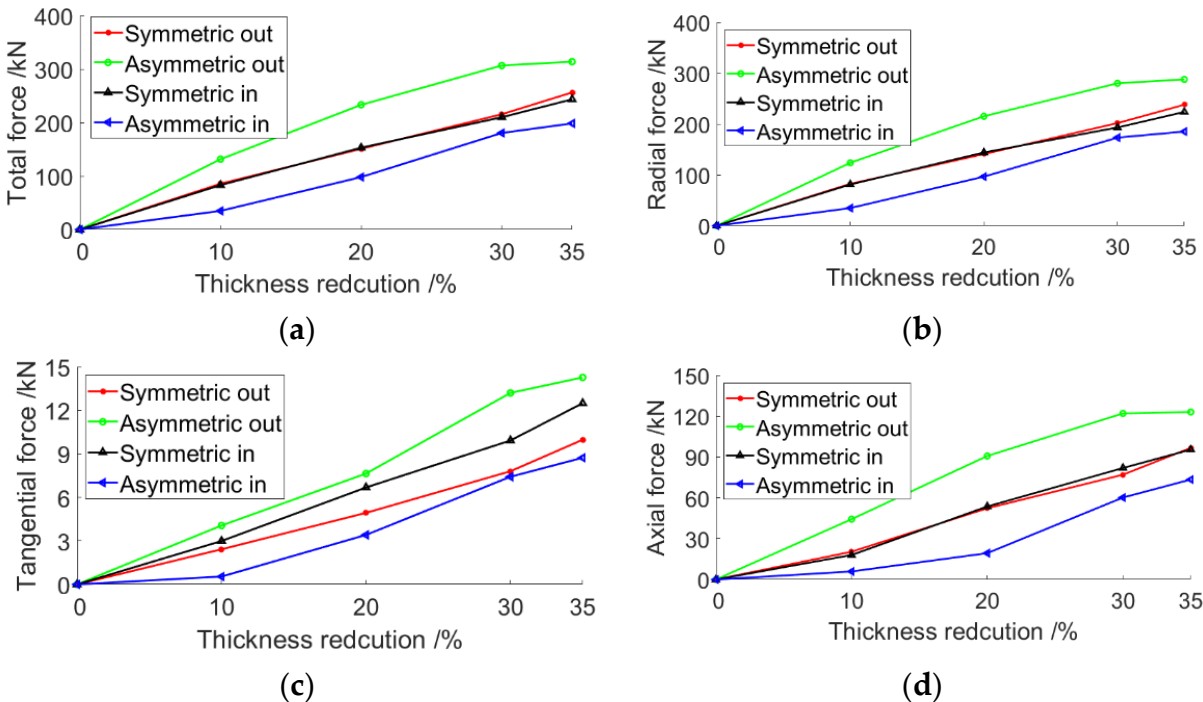

**Figure 18.** The thickness reduction of the tube blank influence on the flow-forming force: (**a**) Total force; (**b**) Radial force; (**c**) Tangential force; (**d**) Axial force. (out: outer roller force; in: inner roller force).

### 4.3.3. Tube Shape

The middle radius and thickness of the final tube in different tube thickness reductions are shown in Figure 19. Both ideal and real middle radii in the SCRFF have a comparable value at a constant of 345 mm. The ideal and real middle radii in the ACRFF are also approximate. The middle radius decreases with the increase of thickness reduction, as shown in Equation (19). Only the real and ideal middle radii with 35% thickness reduction have an observed difference of 1.48 mm or 0.43%. The final tube thickness in both SCRFF and ACRFF processes is approximate to the ideal thickness with a relative difference of <2.8%. Therefore, the thickness reduction nearly has no meaningful influence on the accuracy of both SCRFF and ACRFF final tube thickness.

$$r_{m1} = c_r \cdot t_0 + r_{m0} = -0.5\varnothing \cdot t_0 + r_{m0} \qquad (19)$$

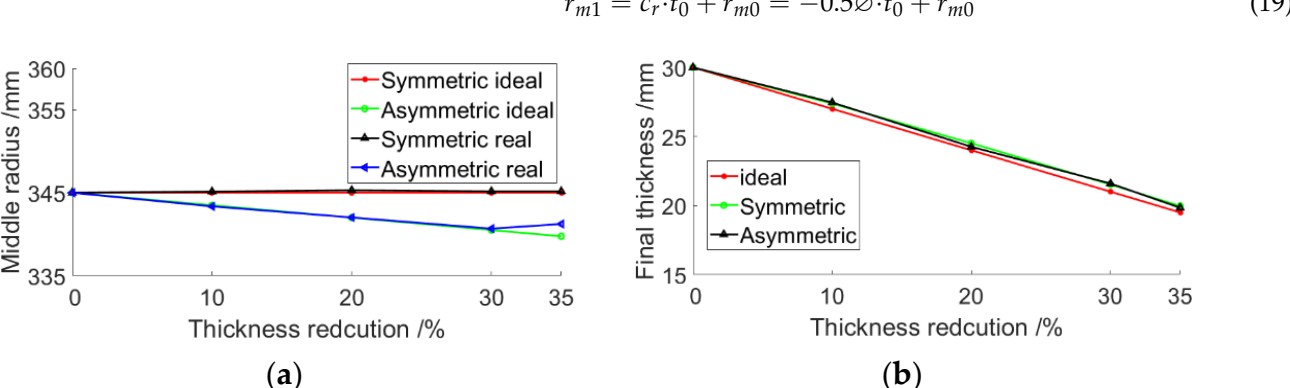

(**a**)　　　　　　　　　　　　　　　　　　　　　　　　(**b**)

**Figure 19.** The thickness of the tube blank influence on the final tube shape: (**a**) Middle radius; (**b**) Thickness (ideal: ideal situation; real: real situation).

### 5. Conclusions

The novel asymmetric counter-roller flow-forming process was introduced and explored. Three main parameters (i.e., tube blank thickness, rollers' offset position, and thickness reduction) were studied by experimental and numerical simulation in the ACRFF process with the following results.

1.  AA5052 aluminum alloy tube can be made by the ACRFF process using a small rollers' offset position (−17.5 to −0%). The main material deformation in this stable forming process is in the axial and radial directions.
2.  The contact line of the outer tube as a typical deformation area features in the ACRFF process increases with the increase of tube blank thickness, the increase of thickness reduction, and the decrease of rollers' offset position. The contact line ratio of the outer and inner tube, which respect the asymmetric deformation, increases with the absolute value of the rollers' offset position. The large contact line ratio can be 4.19.
3.  The roller force difference ($\Delta F$) in the ACRFF process can be more significant than 96 kN. The force difference increases with the increase of tube blank thickness, thickness reduction, and the decrease of rollers' offset position. The inner roller force in the ACRFF process can be 38.6 kN smaller than that in the general CRFF process. The small inner roller force helps reduce the design, construction difficulty, and cost of large tube CRFF equipment.
4.  The difference between the middle diameters in the simulation model and the ideal situation is less than 0.43%. The middle radii of the tube blank and final tube can be 345 mm and 341 mm, respectively. Therefore, the ACRFF can regulate the tube size to improve the flexibility of the CRFF process by changing the rollers' offset position.

**Author Contributions:** Conceptualization, C.Z.; software, D.M.; formal analysis, D.M.; investigation, C.Z. and J.L.; resources, Y.D.; data curation, F.L.; writing—original draft preparation, C.Z.; writing—review and editing, Y.D. and J.L.; visualization, F.L.; project administration, S.Z.; funding acquisition, S.Z. All authors have read and agreed to the published version of the manuscript.

**Funding:** This research was funded by Aerospace Advanced Manufacturing Technology Research Key Program Grant Number U1937203; the Natural Science Foundation of Shaanxi Province, China: 2021JQ-278 and 2021JQ-250; Key Research and Development Program of Shaanxi Province, China: 2019ZDLGY15-01 and 2020ZDZX06-01-01; The open project of Key Laboratory of Education Ministry for Modern Design and Rotor-Bearing System, China.

**Institutional Review Board Statement:** Not applicable.

**Informed Consent Statement:** Not applicable.

**Data Availability Statement:** The data that support the findings of this study are available from the corresponding author upon reasonable request.

**Conflicts of Interest:** The authors declare no conflict of interest. The funders had no role in the design of the study, in the collection, analyses, or interpretation of data, in the writing of the manuscript, or in the decision to publish the results.

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
