# Peer review of "The Rollers’ Offset Position Influence on the Counter-Roller Flow-Forming Process"

_metals, doi:10.3390/met12091471_

Round 1

Reviewer 1 Report

Thank you for the submission of your paper on symmetric and asymetric flow-forming.  Overall, the paper presents a thorough experimental and modeling overview comparison of the two methods of large-tube forming.  A few minor editorial issues should be addressed.

1. Each figure presents experimental or model results and the reader can view each figure to determine the ranges of values presented.  In the text, you can highlight the most important points about each figure. However, it is not necessary to discuss the quantitative values of each parameter in the figures.  In general, the text can be shortened to remove much of the redundant discussion of some of the figures.  Only the major highlights are needed in the text.

2. The word "semblable", page 2, line 56, is not commonly used in English.  Just say "the same".

3. "Intron" should be "Instron".  page 3, line 108.

4. On page 3, line 111, do you mean "true stress-strain"?  Also, in Figure 2, the word "strain" is misspelled.

5.  In Figure 10, there does not appear to be much difference in the magnitude of stress developed for the different cases.  The main difference appears to be only in the length of contact with the rollers.  This should be emphasized in the text.

6. Page 10, line 315:  "liter" should be "little"?

7.  In several places in the paper, the term "error" is used when I think the authors mean there is a "difference" between two values.  When two values are different because of an intentional process parameter change, I think you should use the term "difference".  When discussing an UN-INTENTIONAL variation in a measured values, you should use the term "error".  Error would be similar to "scatter" in results.

8. Page 13, lines 424-425.  It is a little confusing when saying the "middle radius increases with the increase of the rollers' offset position".  Yes, the radius increases as the offset position is moved to "less negative" values towards zero the symmetrc case.  However, the radius DECREASES as the offset increases (in a negative direction).  The wording here can be modified to make it clear that the radius decreases as the rollers become more offset from each other.

9. Again, in CONCLUSION 3, the use of the word "error" should be changed to "difference"

Author Response

Dear Professor,

Thanks for the valuable and insightful comments and the opportunity to revise and resubmit this manuscript. We appreciated the constructive criticism of the reviewers and have answered these and incorporated the suggestions into the manuscript where necessary. The manuscript has definitely benefited from the insightful revision advice. We look forward to working with you and the reviewers to move this manuscript closer to publication with Metals.

We have modified the manuscript accordingly, and detailed corrections are listed below point by point. To make the response and changes easier to identify where necessary, the reviewers’ comments are presented with the response from author is presented in blue, and the corresponding corrections in the manuscript are presented in red.

Manuscript ID: metals-1863735

Title: The rollers’ offset position influence on the counter-roller flow-forming process

Journal: Metals (ISSN 2075-4701)

Comments and Suggestions for Authors:

“Thank you for the submission of your paper on symmetric and asymetric flow-forming.  Overall, the paper presents a thorough experimental and modeling overview comparison of the two methods of large-tube forming.  A few minor editorial issues should be addressed.”

Reviewer #1: Each figure presents experimental or model results and the reader can view each figure to determine the ranges of values presented.  In the text, you can highlight the most important points about each figure. However, it is not necessary to discuss the quantitative values of each parameter in the figures.  In general, the text can be shortened to remove much of the redundant discussion of some of the figures.  Only the major highlights are needed in the text.

Response to Reviewer #1 comment No. 1:

Thanks very much for your comment. The large paragraphs before Figure 6, Figure 8, Figure9, Figure 11, Figure 14, Figure 16, and Figure 18 have been simplified to highlight the essential points.

For example, the paragraph before Figure 8 was simplified.“The thickness of the tube blank was 10 mm. The stage tube made by the ACRFF process with 10%, 20%, and 30% ideal thickness reduction is shown in Figure 8. The final tube shape in numerical simulation and experiment are similar. The final tube shape values are shown in Table 1. In the 10% thickness reduction ACRFF process, the tube form depth difference and the rollers’ offset position are −0.32 and −0.16 mm, respectively. In the 20% thickness reduction ACRFF process., the tube form depth difference and the rollers’ offset position are −0.32 and −0.16 mm, respectively. In the 30% thickness reduction ACRFF process, the form depth difference and the rollers’ offset position are −1.02 and −0.51 mm, respectively. The middle radius of the tube blank and the final tube in the general SCRFF process is 355 mm. The rollers’ offset position changed the middle radii of the final tubes to 354.84 and 354.89 mm. Therefore, the middle radius of the CRFF tube depends on the rollers’ offset position.”

  1. The word "semblable", page 2, line 56, is not commonly used in English.  Just say "the same".

Response to Reviewer #1 comment No. 2:

Thanks for your comment. The word “semblable” has been changed to “the same”.

  1. "Intron" should be "Instron".  page 3, line 108.

Response to Reviewer #1 comment No. 3:

Thanks for your comment, it’s my mistake. The word “intron” has been changed to “Instron”.

  1. On page 3, line 111, do you mean "true stress-strain"? Also, in Figure 2, the word "strain" is misspelled.

Response to Reviewer #1 comment No. 4:

Thanks for your comment, it’s my mistake. The "real stress-strain" has been changed to "true stress-strain". The word “strain” in Figure 2(b) has been changed.

Figure 2. Tensile samples and strain-stress curve of AA5052 aluminum alloy: (b) Strain-stress curve.

  1. In Figure 10, there does not appear to be much difference in the magnitude of stress developed for the different cases. The main difference appears to be only in the length of contact with the rollers. This should be emphasized in the text.

Response to Reviewer #1 comment No. 5:

Thanks for your comment. The introduction of the phenomena was added into the paragraph. “The tube blank material was AA5052 aluminum alloy whose hardening curve has a prominent saturation feature in significant deformation situations. The strain hardening curve used in the ABAQUS is Equation (8). The maximum plastic strain used in this mate-rial model is 0.25 as the fracture strain in the tensile test. When the plastic strain is larger than 0.25, the stress will be an invariable value. Since the Equivalent plastic strain (PEEQ) in all numerical simulations is larger than 0.25, the stress will reach the large invariant value in the material model. Furthermore, there are stress concentration phenomena in the contact area between the roller and the tube blank for large deformation.”

,

(8)

  1. Page 10, line 315: "liter" should be "little"?

Response to Reviewer #1 comment No. 6:

Thanks for your comment, it’s my mistake. The word “liter” has been changed to “little”.

  1. In several places in the paper, the term "error" is used when I think the authors mean there is a "difference" between two values. When two values are different because of an intentional process parameter change, I think you should use the term "difference". When discussing an UN-INTENTIONAL variation in a measured values, you should use the term "error". Error would be similar to "scatter" in results.

Response to Reviewer #1 comment No. 7:

Thanks very much for your comment. It’s my mistake. The number of the word “error” is 37 in the paper. I have replaced all the word “error” by “difference”. The Figure 14 has also been changed.

(a)                                    (b) 

(c)                                    (d) 

Figure 14. The influence of the rollers’ offset position on the flow-forming force: (a) Total force; (b) Radial force; (c) Tangential force; (d) Axial force. (Mean: mean force; Difference: force difference.)

  1. Page 13, lines 424-425. It is a little confusing when saying the "middle radius increases with the increase of the rollers' offset position". Yes, the radius increases as the offset position is moved to "less negative" values towards zero the symmetrc case.  However, the radius DECREASES as the offset increases (in a negative direction).  The wording here can be modified to make it clear that the radius decreases as the rollers become more offset from each other.

Response to Reviewer #1 comment No. 8:

Thanks very much for your comment. This sentence “The middle radius increases with the increase of the rollers’ offset position value” has been changed.

The new sentence is:

“Since the rollers’ offset value is in the negative direction in this study, the rollers’ offset value increases with its absolute value decreasing. The middle radius increases with the increase of the rollers’ offset position value in this situation. When the roller’s offset position is negative, the middle radius increases with the absolute value of the rollers’ offset position decreasing.”

This sentence has also been write in the Section 4.2.1. “Since the rollers’ offset value is in the negative direction in this study, the rollers’ offset value increases with its absolute value decreasing.”

  1. Again, in CONCLUSION 3, the use of the word "error" should be changed to "difference".

Response to Reviewer #1 comment No. 9:

Thanks for your comment. The word "error" has been changed.

Best regards

Chengcheng Zhu

Reviewer 2 Report

The authors present the analysis of the asymmetric mating roll flow-forming process in which three main parameters were investigated: the tube thickness, the staggered position of the rolls and the thickness reduction ratio.

General remarks.
1.) The paper seems to lack a clear description of the novelty of the work. This should be made clear in the abstract or in the introduction of the paper. It is somehow unclear what motivates the study and what the research gap is. The research gap and the novelty should be clearly pointed out.
2.) The paper is more of an industrial case study than a research paper. The conclusions do not offer any new insights into the process. Rather, they all appear logical and without numerical analysis. The introduction and conclusion should therefore be improved.
3.) From the stress contour plots in Figure 10, it appears that your numerical model is inadequate because the same stress level is reached and the von Mises stress remains constant while the plastic strain evolves.
First, please give the values of the strain hardening curve given in ABAQUS. Up to what value did you specify the yield curve? Up to PEEQ=0.3? Did you extrapolate the hardening curve? How was your strain hardening curve extrapolated, since the strains in your process are definitely higher than those characterised by the uniaxial test? For different extrapolations of the strain hardening law, see e.g. 10.1016/j.jmatprotec.2019.03.010. Please indicate the maximum value of the PEEQ achieved in the simulation.

Specific comments:
- Line 106: "The elastoplastic constitutive model of aluminium alloy AA5051 was obtained by the uniaxial test... " is somewhat misleading. You have only given a yield curve. The elastoplastic constitutive model is more than just a yield curve, it includes the yield criterion, the hardening law, the addition of plastic and elastic strains, the associative plastic flow rule, Hooke's law and the equivalence of plastic work (see e.g. 10.1016/j.euromechsol.2013.11.013).
- Line 110: "The strain rate was considered constant because the strain rate of the sample is low". At first glance, this seems to be a recursive statement and also contradictory. With the hardening law (8) you have stated that the yield stress follows a power law. Please reformulate the statement.
- Line 116,119: Typing error
- Figure 2 does not follow the relationship (8). You have also drawn a section of the curve that corresponds to the fracture, although the fracture has not been modelled in ABAQUS.
- Lines 141-145: Monotone English language, please improve.
- Units (MPa) should be given for different stress contour diagrams

Author Response

Dear Professor,

Thanks for the valuable and insightful comments and the opportunity to revise and resubmit this manuscript. We appreciated the constructive criticism of the reviewers and have answered these and incorporated the suggestions into the manuscript where necessary. The manuscript has definitely benefited from the insightful revision advice. We look forward to working with you and the reviewers to move this manuscript closer to publication with Metals.

We have modified the manuscript accordingly, and detailed corrections are listed below point by point. To make the response and changes easier to identify where necessary, the reviewers’ comments are presented with the response from author is presented in blue, and the corresponding corrections in the manuscript are presented in red.

Manuscript ID: metals-1863735

Title: The rollers’ offset position influence on the counter-roller flow-forming process

Journal: Metals (ISSN 2075-4701)

Comments and Suggestions for Authors:

“The authors present the analysis of the asymmetric mating roll flow-forming process in which three main parameters were investigated: the tube thickness, the staggered position of the rolls and the thickness reduction ratio.”

Reviewer #1: The paper seems to lack a clear description of the novelty of the work. This should be made clear in the abstract or in the introduction of the paper. It is somehow unclear what motivates the study and what the research gap is. The research gap and the novelty should be clearly pointed out.

Response to Reviewer #1 comment No. 1:

Thanks very much for your comment. The “research gap and the novelty” have been written in the abstract and introduction. 

“The general counter-roller flow-forming (CRFF) process rarely considers the roller’s offset position for the symmetric rollers. However, the rollers’ offset position could regulate the tube shape, force and other features. Studying the novel asymmetric CRFF process, which is the CRFF process with the rollers’ offset position is essential.” “AA5052 aluminum alloy tube can be made by the novel asymmetric CRFF process using a small rollers’ offset position (−17.5%–0%).”

“Thus, there are rare studies on the roller’s offset position in the CRFF process. However, the rollers’ position difference usually occurs in the actual flow-forming device, which can significantly change the tube shape, flow-forming force, and other features. Furthermore, we can control the rollers’ position to improve the CRFF flexibility and obtain various tubes with the same tube blanks. The rollers’ position difference is the rollers’ offset position when using the same rollers. Therefore, the influence of the rollers’ offset position on the CRFF process needs to be studied. This novel CRFF process in which we actively control rollers’ offset position is called the asymmetrical CRFF process.”

“This study is mainly about the rollers’ offset position in the novel ACRFF process by numerical simulation and experimental methods.” “Various sizes of AA5052 aluminum alloy tubes can be made by the novel ACRFF process, which improves the flexibility of the CRFF process. The small inner roller force in the ACRFF helps improve the capability and simplify the structure of CRFF equipment. This study can improve the ACRFF deformation mechanism recognition and various large-tube manufacturing methods.”

  1. The paper is more of an industrial case study than a research paper. The conclusions do not offer any new insights into the process. Rather, they all appear logical and without numerical analysis. The introduction and conclusion should therefore be improved.

Response to Reviewer #1 comment No. 2:

Thanks very much for your comment. The introduction and conclusion have been rewritten.

Conclusion:

“The novel asymmetric counter-roller flow-forming process was raised and explored. Three main parameters (i.e., tube blank thickness, rollers’ offset position, and thickness reduction) were studied by experimental and numerical simulation in the ACRFF process.

  1. AA5052 aluminum alloy tube can be made by the ACRFF process using a small rollers’ offset position (−17.5% to −0%). The main material deformation in this stable forming process is in the axial and radial directions.
  2. The contact line of the outer tube as a typical deformation area features in the ACRFF process increases with the increase of tube blank thickness, the increase of thickness reduction, and the decrease of rollers’ offset position. The contact line ratio of the outer and inner tube, which respect the asymmetric deformation, increases with the absolute value of rollers’ offset position. The large contact line ratio can be 4.19.
  3. The roller force difference ( ) in the ACRFF process can be more significant than 96 kN. The force difference increases with the increase of tube blank thickness, thickness reduction, and the decrease of rollers’ offset position. The inner roller force in the ACRFF process can be 38.6 kN smaller than that in the general CRFF process. The small inner roller force helps reduce the design, construction difficulty, and cost of large tube CRFF equipment.
  4. The difference between the middle diameters in the simulation model and the ideal situation is less than 0.43%. The middle radii of the tube blank and final tube are 345 mm and 341mm, respectively. Therefore, the ACRFF can regulate the tube size to improve the flexibility of the CRFF process by changing the rollers’ offset position. ”

Introduction:

“This study is mainly about the rollers’ offset position in the novel ACRFF process by numerical simulation and experimental methods. The AA5052 aluminum alloy tube was the research target. The influences of the three significant parameters are the tube blank thickness, thickness reduction, and rollers’ offset position on the material deformation, flow-forming force, the middle radius, and the final tube thickness. Various sizes of AA5052 aluminum alloy tubes can be made by the novel ACRFF process, which improves the flexibility of the CRFF process. The small inner roller force in the ACRFF helps improve the capability and simplify the structure of CRFF equipment. This study can improve the ACRFF deformation mechanism recognition and various large-tube manufacturing methods.”

  1. From the stress contour plots in Figure 10, it appears that your numerical model is inadequate because the same stress level is reached and the von Mises stress remains constant while the plastic strain evolves.

First, please give the values of the strain hardening curve given in ABAQUS. Up to what value did you specify the yield curve? Up to PEEQ=0.3? Did you extrapolate the hardening curve? How was your strain hardening curve extrapolated, since the strains in your process are definitely higher than those characterised by the uniaxial test? For different extrapolations of the strain hardening law, see e.g. 10.1016/j.jmatprotec.2019.03.010. Please indicate the maximum value of the PEEQ achieved in the simulation.

Response to Reviewer #1 comment No. 3:

Thanks very much for your comment. The relative information has been written into the paper.

There are many methods to extrapolate the hardening curve[1], but it is challenging to choose an appreciative one. Since our purpose is the novel ACRFF process, the material model used in the ABAQUS may not be perfect. However, the hardening curve is enough to study the forming process's primary feature. The high accuracy material hardening curve will be researched in further study.

“The strain hardening curve used in the ABAQUS is Equation 8. The maximum plastic strain used in this material model is 0.25 as the fracture strain in the tensile test. The stress will be an invariable value when the plastic strain (PEEQ) is larger than 0.25.” This is the method that we used to extrapolate the hardening curve.

,

(8)

“The largest PEEQ value in the contact area in numerical simulation is 1.7, which is bigger than the equivalent strain of 0.5 in the theoretical model. The reason is the asymmetric deformation and cyclic loading in the ACRFF process.”  

The reasons for using this hardening curve are:

The compression test of the AA5052 aluminum alloy was also done. The true strain-stress curve is shown in Figure 1. The AA5052 aluminum alloy hardening curve has a prominent saturation feature. When the strain is larger than 0.2, the stress is nearly invariant. This is why I haven’t used Equation (8) when the strain is significant.

Generally, the tensile test has higher accuracy than the compression test, which has friction and minor sample problems. Thus, the tensile test was chosen to build the hardening curve. Since the AA5052 aluminum alloy fracture strain in uniaxial tensile is much smaller than in the compression test and counter-roller flow-forming process (also at the compression stress state), the hardening model is mixed. When the true strain is smaller than 0.25, the strain hardening curve used in the ABAQUS is Equation (8). When the true strain is larger than 0.25, the stress will be an invariable value for the saturation feature.

Figure1. The AA5052 aluminum alloy true strain-stress cure in the compression test

Reference

[1] Starman, B., et al. “Shear test-based identification of hardening behaviour of stainless steel sheet after onset of necking.” Journal of Materials Processing Technology 270 (2019): 335-344.

  1. Line 106: “The elastoplastic constitutive model of aluminium alloy AA5051 was obtained by the uniaxial test...” is somewhat misleading. You have only given a yield curve. The elastoplastic constitutive model is more than just a yield curve, it includes the yield criterion, the hardening law, the addition of plastic and elastic strains, the associative plastic flow rule, Hooke’s law and the equivalence of plastic work (see e.g. 10.1016/j.euromechsol.2013.11.013).

Response to Reviewer #1 comment No. 4:

Thanks very much for your comment. It’s my mistake. This sentence has been rewritten. “The AA5052 aluminum alloy elastoplastic constitutive model can be obtained by the Mises yield criterion, isotropic hardening model, linear elastic theory and the associated flow rule. The true strain-stress curve was fitted by the power exponent model shown in Equation (8), whose R-value is bigger than 99% and mean square error is less than 8 MPa.”

  1. Line 110: “The strain rate was considered constant because the strain rate of the sample is low”. At first glance, this seems to be a recursive statement and also contradictory. With the hardening law (8) you have stated that the yield stress follows a power law. Please reformulate the statement.

Response to Reviewer #1 comment No. 5:

Thanks very much for your comment. This sentence has been rewritten. “The strain rate of the sample during the test slightly decreased from 0.001 s-1 to 0.00076 s-1, so it is feasible to assume that the strain rate in the test is approximate invariant.”

  1. Line 116,119: Typing error.

Response to Reviewer #1 comment No. 6:

Thanks for your comment.

These errors have been corrected.

Figure 2. Tensile samples and strain-stress curve of AA5052 aluminum alloy: (b) Strain-stress curve.

  1. Figure 2 does not follow the relationship (8). You have also drawn a section of the curve that corresponds to the fracture, although the fracture has not been modelled in ABAQUS.

Response to Reviewer #1 comment No. 7:

Thanks very much for your comment. It’s my mistake. Figure 2 was changed.

Figure 2. Tensile samples and strain-stress curve of AA5052 aluminum alloy: (b) Strain-stress curve.

  1. Lines 141-145: Monotone English language, please improve.

Response to Reviewer #1 comment No. 8:

Thanks for your comment. These sentences have been changed.

Original: “The CRFF experiment was done on a numerical control CRFF device (Figure 4). The tube blank was made of AA5052 aluminum alloy. The outer diameter and thickness of the tube blank were 720 and 10 mm, respectively. The back-forward CRFF experiment was done at room temperature. The tube blank bottom was fixed on the turntable. The rotation speed of the turntable is 40 r/min.”

New: “The CRFF experiment was done on a numerical control CRFF device (Figure 4) at room temperature. The outer diameter and thickness of the AA5052 aluminum alloy tube blank were 720 and 10 mm, respectively. Since the experiment was based on the back-forward CRFF process, the tube blank bottom was fixed on the turntable. The turntable rotation and roller feed speed are 0.67 r/s and 1 mm/s, respectively.”

  1. Units (MPa) should be given for different stress contour diagrams.

Response to Reviewer #1 comment No. 9:

Thanks for your comment. Figure 7(a), Figure 10, Figure 16 and Figure 17 have been changed. The Units (MPa) has been given in these figures.

          (a)                             (b) 

Figure 7. The stress nephogram of the CRFF part: (a) Total model; (b) Final tube net.

(a)                        (b)                         (c)

(d)                      (e)                           (f)

Figure 10. The stress nephograms of the tube deformation region in the CRFF process: (a) 10 mm thickness, ACRFF; (b) 20 mm thickness, ACRFF; (c) 30 mm thickness, ACRFF; (d) 10 mm thickness, SCRFF; (e) 20 mm thickness, SCRFF; (f) 30 mm thickness, SCRFF.

(a)                                    (b) 

(c)                                    (d) 

Figure 16. The stress nephograms of the tube deformation region with different thickness reduction in ACRFF processes: (a) 10%; (b) 20%; (c) 30%; (d) 35%.

Figure 17. The stress nephogram of the tube deformation region in 10% thickness reduction SCRFF process.

Best regards

Chengcheng Zhu
